# Fast multi-source nanophotonic simulations using augmented partial factorization

Ho-Chun Lin ⬡ , Zeyu Wang & Chia Wei Hsu ⬡ ✉

Numerical solutions of Maxwell's equations are indispensable for nanophotonics and electromagnetics but are constrained when it comes to large systems, especially multi-channel ones such as disordered media, aperiodic metasurfaces and densely packed photonic circuits where the many inputs require many large-scale simulations. Conventionally, before extracting the quantities of interest, Maxwell's equations are first solved on every element of a discretization basis set that contains much more information than is typically needed. Furthermore, such simulations are often performed one input at a time, which can be slow and repetitive. Here we propose to bypass the full-basis solutions and directly compute the quantities of interest while also eliminating the repetition over inputs. We do so by augmenting the Maxwell operator with all the input source profiles and all the output projection profiles, followed by a single partial factorization that yields the entire generalized scattering matrix via the Schur complement, with no approximation beyond discretization. This method applies to any linear partial differential equation. Benchmarks show that this approach is 1,000–30,000,000 times faster than existing methods for two-dimensional systems with about 10,000,000 variables. As examples, we demonstrate simulations of entangled photon backscattering from disorder and high-numerical-aperture metalenses that are thousands of wavelengths wide.

The interaction between light and nanostructured materials leads to rich properties. For small systems such as individual nano/microstructures and optical components, or for periodic systems such as photonic crystals and periodic metamaterials, one can readily solve Maxwell's equations numerically to obtain predictions that agree quantitatively with experiments. However, the computational costs are typically too heavy for more complex systems such as disordered ones[1] that not only are large but also couple many incoming channels to many outgoing ones, requiring numerous simulations. The alternatives all have limitations: the Born approximation does not describe multiple scattering, radiative transport and diagrammatic methods can only compute some ensemble-averaged properties[2] and coupled-mode theory requires systems with isolated resonances[3,4]. For metasurfaces[5], the widely used locally periodic approximation[5,6] is inaccurate whenever the cell-to-cell variation is large[7–9] and cannot describe nonlocal responses[10] or metasurfaces that are not based on unit cells[11,12].

Classical and quantum photonic circuits build on individual components that couple very few channels at a time, limiting the number of inputs and outputs. Examples beyond photonics also abound. A wide range of studies across different disciplines are currently prohibited by computational limitations.

Regardless of the complexity of a system, its linear response is described exactly by an $M' \times M$ generalized scattering matrix **S** that relates an arbitrary input vector **v** to the resulting output vector **u** via[13,14]

$$u_n = \sum_{m=1}^{M} S_{nm} v_m. \tag{1}$$

The $M$ columns of **S** correspond to $M$ distinct inputs (Fig. 1a,b), which can be different incoming angles or beam profiles, different waveguide modes, different point dipole excitations, their superpositions or any

Ming Hsieh Department of Electrical and Computer Engineering, University of Southern California, Los Angeles, CA, USA. ✉e-mail: cwhsu@usc.edu

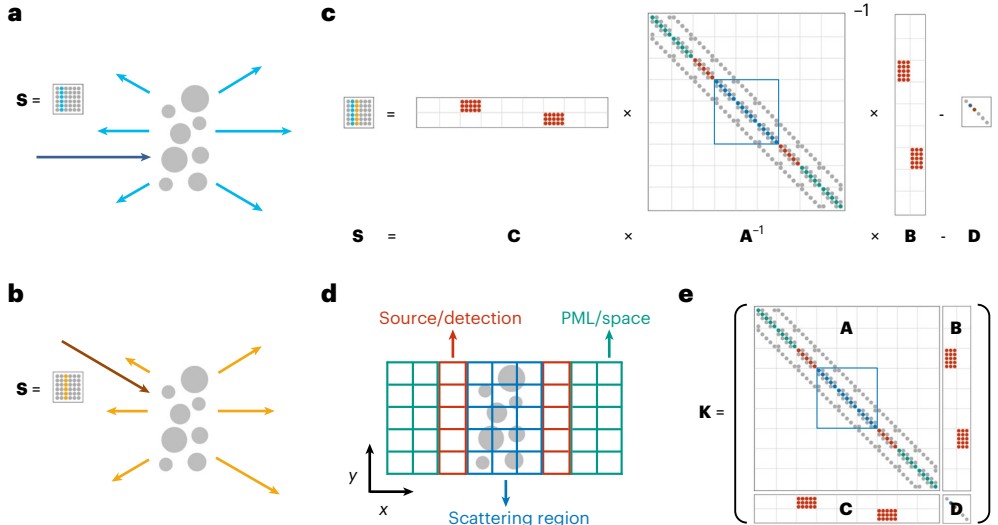

**Fig. 1 | Generalized scattering matrix and augmented partial factorization (APF). a,b,** Schematic of light scattering from a nanostructure (grey circles). Outgoing waves (light-blue and orange arrows) resulting from inputs at two different incident angles (dark-blue and brown arrows) correspond to two different columns of the scattering matrix **S** (insets). **c,** Illustration of equation (2), which relates a generalized scattering matrix **S** to the inverse of the discretized Maxwell operator **A**, source profiles **B** that generate the incident waves, projection profiles **C** that extract the outputs of interest and the matrix **D** that subtracts the baseline. Each small circle indicates a nonzero element of the sparse matrix, coloured based on its spatial location shown in **d. d,** Discretization grids of the illustration in **c,** with colour coding for different regions of the system. **e,** The augmented sparse matrix **K** of equation (3), whose partial factorization gives the generalized scattering matrix **S**.

other input of interest. Similarly, the vector **u** can contain any output of interest in the near field or far field.

Computing such a multi-input response typically requires $M$ distinct solutions of Maxwell's equations with the same structure given different source profiles. Time-domain methods[15] are easy to parallelize but cannot leverage the multi-input property. Frequency-domain methods allow strategies for handling many inputs. After volume discretization onto a basis through finite element[16] or finite difference[17], Maxwell's equations in the frequency domain become a system of linear equations $\mathbf{A}\mathbf{x}_m = \mathbf{b}_m$. The sparse matrix **A** is the Maxwell differential operator, the column vector $\mathbf{b}_m$ on the right-hand side specifies the $m$th input and the full-basis solution is contained in the column vector $\mathbf{x}_m$. When solving for $\mathbf{x}_m = \mathbf{A}^{-1}\mathbf{b}_m$ using direct methods, the sparsity can be utilized via graph partitioning, and the resulting lower–upper (LU) factors can be reused among different inputs[18,19]. However, $M$ forward and backward substitutions are still needed, and the LU factors take up substantial memory. Iterative methods compute $\mathbf{x}_m = \mathbf{A}^{-1}\mathbf{b}_m$ by minimizing the residual[20], avoiding the LU factors. One can iterate multiple inputs together[21] or construct preconditioners to be reused among different inputs[22,23], but the iterations still take $\mathcal{O}(M)$ time.

For homogeneous structures with small surface-to-volume ratio, the boundary element method[24] can efficiently discretize the interface between materials to reduce the size and the condition number of the matrix **A**, though its matrix **A** is no longer sparse. Instead of a surface mesh, the $T$-matrix method[25] uses vector spherical harmonics as basis functions, also resulting in a dense matrix **A**. The hierarchical structure of the dense matrix **A** can be utilized through the fast multipole method[26] within iterative solvers or through the $\mathcal{H}$-matrix method[27] within direct solvers, but the computing time still scales as $\mathcal{O}(M)$.

For systems with a closed boundary on the sides and inputs/outputs placed on the front and back surfaces, the recursive Green's function (RGF) method[28] can obtain the full scattering matrix without looping over the inputs, which is useful for disordered systems[1]. However, the RGF method works with dense Green's function matrices and thus scales unfavourably with the system width $W$ as $\mathcal{O}(W^{3(d-1)})$ for computing time and $\mathcal{O}(W^{2(d-1)})$ for memory usage in $d$ dimensions. For layered geometries, the rigorous coupled-wave analysis (RCWA)[29] and the eigenmode expansion[30] methods use local eigenmodes to utilize the intralayer axial translational symmetry, which also results in dense matrices and the same scaling as the RGF method.

All these methods solve Maxwell's equations on every element of the discretization basis set, typically one input at a time, after which the quantities of interest are extracted from the solutions. Doing so is intuitive but leads to unnecessary computations and repetitions. Here, we propose the augmented partial factorization (APF) method that directly computes the entire generalized scattering matrix of interest, bypassing the full-basis solutions and without repeating over the inputs. APF is general (applicable to any structure with any type of inputs and outputs, including to other linear partial differential equations), exact (no approximation beyond discretization), does not store large LU factors, scales well with the system size and fully utilizes the sparsities of the Maxwell operator, of the inputs and also of the outputs. These advantages lead to reduced memory usage and a speed-up of many orders of magnitude compared with existing methods (even those that specialize in a certain geometry), enabling full-wave simulations of massively multi-channel systems that were impossible in the past.

## Results

### Augmented partial factorization

Regardless of the discretization scheme (finite difference, finite element, boundary element, $T$-matrix, spectral methods, etc.), a frequency-domain simulation for the $m$th input reduces to computing $\mathbf{x}_m = \mathbf{A}^{-1}\mathbf{b}_m$. Considering $M$ inputs, the collective full-basis solutions are $\mathbf{X} = \mathbf{A}^{-1}\mathbf{B}$ where $\mathbf{X} = [\mathbf{x}_1, \ldots, \mathbf{x}_M]$ and $\mathbf{B} = [\mathbf{b}_1, \ldots, \mathbf{b}_M]$. The full content of this dense and large matrix **X** is rarely needed. The needed quantities are encapsulated in the generalized scattering matrix **S**, which we can write as

$$\mathbf{S} = \mathbf{C}\mathbf{A}^{-1}\mathbf{B} - \mathbf{D}. \qquad (2)$$

The matrix **C** projects the collective solutions $\mathbf{X} = \mathbf{A}^{-1}\mathbf{B}$ onto the $M'$ outputs of interest (for example, sampling at the locations of interest, a conversion to propagating channels or a transformation from the near field to far field[15]). It is sparse since the projections only use part

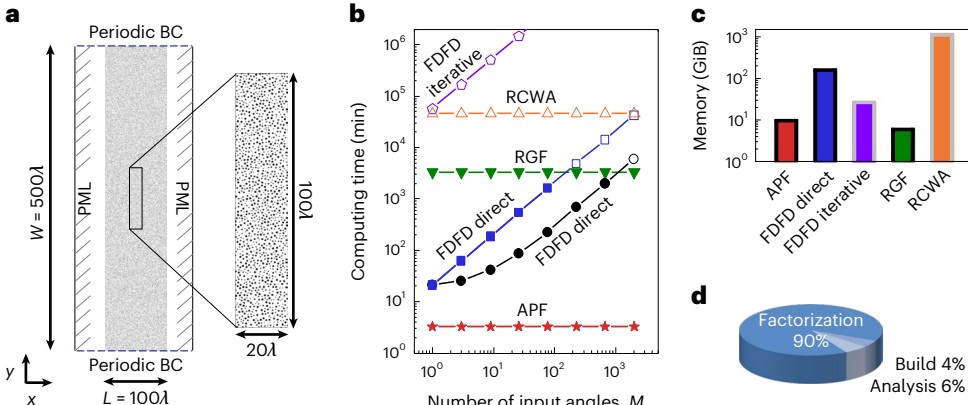

**Fig. 2 | Benchmarks on a large-scale disordered system. a**, The system considered consists of 30,000 randomly positioned cylindrical scatterers in air, each with refractive index of 2.0 and diameter between $0.3\lambda$ and $0.8\lambda$, where $\lambda$ is the wavelength. A periodic boundary condition is used in the $y$ direction, and perfectly matched layers (PMLs) are used in the $\pm x$ directions as outgoing boundaries. We compute the scattering matrix with up to $2W/\lambda = 1,000$ plane-wave inputs from either the left or right and with all of the $M' = 2,000$ outgoing plane waves. **b**, Computing time versus the number $M$ of input angles using APF and other methods: conventional FDFD method using MaxwellFDFD

with direct[40] or iterative[41] solvers for the full-basis solutions, RCWA using S4 (ref. [43]) and the RGF method[42]. Open symbols are extrapolated from smaller $M$ or smaller systems. The two 'FDFD direct' curves correspond to an unmodified version of MaxwellFDFD (blue squares) and one modified to have the LU factors reused for different inputs (black circles). **c**, Memory usage of different methods; grey-edged bars are extrapolated from smaller systems. **d**, Breakdown of the APF computing time into time used in building the matrix **K**, analysing and reordering it, and partially factorizing it.

---

of the solutions, and it is very fat since the number $M'$ of outputs of interest is generally far less than the number of discretization basis elements. The matrix $\mathbf{D} = \mathbf{C}\mathbf{A}_0^{-1}\mathbf{B} - \mathbf{S}_0$ subtracts the baseline contribution from the incident field (Supplementary Fig. 1), where $\mathbf{A}_0$ is the Maxwell operator of a reference system (for example, vacuum) for which the generalized scattering matrix $\mathbf{S}_0$ is known. This ensures that $\mathbf{S}$ reduces to $\mathbf{S}_0$ when $\mathbf{A}$ becomes $\mathbf{A}_0$. Equation (2) has the same superficial structure as scattering matrices in quasi-normal coupled mode theory[4] but is simpler and does not require the computation of quasi-normal modes (which is expensive for large systems).

Given the generalized scattering matrix $\mathbf{S}$, the response to other inputs can be obtained from superposition, as in equation (1). Time-dependent responses are given by Fourier transforming the frequency-domain response[31].

Figure 1c,d illustrates equation (2) with a concrete example. Consider the transverse magnetic fields in two dimensions (2D) for a system periodic in $y$ with a relative permittivity profile of $\varepsilon_r(\mathbf{r}) = \varepsilon_r(x, y)$. The Maxwell differential operator on the out-of-plane electric field $E_z(\mathbf{r})$ at wavelength $\lambda$ is $-\nabla^2 - (2\pi/\lambda)^2 \varepsilon_r(\mathbf{r})$, which becomes the matrix $\mathbf{A}$ when volume is discretized with an outgoing boundary in the $x$ direction. Then, the matrix $\mathbf{A}^{-1}$ is the retarded Green's function $G(\mathbf{r}, \mathbf{r}')$ of this system. A plane wave incident from the left, $e^{i(k_x^{in}x + k_y^{in}y)}$, can be generated with a source proportional to $\delta(x)e^{ik_y^{in}y}$ on the front surface $x = 0$ where $\delta(x)$ is the Dirac delta function, and incident waves from the right can be similarly generated. These source profiles become the columns of the matrix $\mathbf{B}$ when discretized. The coefficients of different outgoing plane waves to the left can be obtained from projections proportional to $\delta(x)e^{-ik_y^{out}y}$, and similarly with outgoing waves to the right. They become the rows of the matrix $\mathbf{C}$ when discretized. In this particular example, $\mathbf{D} = \mathbf{I}$ is the identity matrix, and equation (2) reduces to the discrete form of the Fisher–Lee relation in quantum transport[32] (Supplementary Sects. 1 and 2 and Supplementary Fig. 1). We only show a few discretized pixels and a few angles in Fig. 1c,d to simplify the schematic. In reality, the numbers of pixels and input angles can readily exceed millions and thousands, respectively. Note that the matrices $\mathbf{A}$, $\mathbf{B}$ and $\mathbf{C}$ are all sparse here.

Instead of solving for $\mathbf{X} = \mathbf{A}^{-1}\mathbf{B}$ as is conventionally done, we directly compute the generalized scattering matrix $\mathbf{S} = \mathbf{C}\mathbf{A}^{-1}\mathbf{B} - \mathbf{D}$, which is orders of magnitude smaller. To do so, we build an augmented sparse matrix $\mathbf{K}$ as illustrated in Fig. 1e and then perform a partial factorization:

$$\mathbf{K} \equiv \begin{bmatrix} \mathbf{A} & \mathbf{B} \\ \mathbf{C} & \mathbf{D} \end{bmatrix} = \begin{bmatrix} \mathbf{L} & \mathbf{0} \\ \mathbf{E} & \mathbf{I} \end{bmatrix} \begin{bmatrix} \mathbf{U} & \mathbf{F} \\ \mathbf{0} & \mathbf{H} \end{bmatrix}. \tag{3}$$

The factorization is partial as it stops after factorizing the upper left block of $\mathbf{K}$ into $\mathbf{A} = \mathbf{L}\mathbf{U}$. Such partial factorization can be carried out using established sparse linear solver packages such as MUMPS[33] and PARDISO[34]. Notably, we do not use the LU factors, and the $\mathbf{L}$ and $\mathbf{U}$ in this APF formalism do not even need to be triangular. By equating the middle and the right-hand side of equation (3) block by block, we see that the matrix $\mathbf{H}$, called the Schur complement[35], satisfies $\mathbf{H} = \mathbf{D} - \mathbf{C}\mathbf{A}^{-1}\mathbf{B}$. Thus, we obtain the generalized scattering matrix via $\mathbf{S} = -\mathbf{H}$. In this way, a single factorization yields what conventional methods obtain from $M$ separate simulations. Repetitions over inputs are no longer necessary. We name this approach augmented partial factorization (APF).

APF is as general as equation (2), applicable to any linear partial differential equation, in any dimension, under any discretization scheme, with any boundary condition, for any type of inputs generated using any scheme (such as equivalent source for arbitrary incident waves like waveguide modes[17,36], line source and point dipole source) and for any type of output projections. As a frequency-domain method, it works with arbitrary material dispersion, and the response at different frequencies can be computed independently. It is a full-wave method as precise as the underlying discretization.

APF avoids a slow loop over the $M$ inputs or a slow evaluation of the dense Green's function. The sparsity patterns of $\mathbf{A}$, $\mathbf{B}$ and $\mathbf{C}$ are maintained in $\mathbf{K}$ and can all be utilized during the partial factorization. The matrices $\mathbf{L}$ and $\mathbf{U}$ are not as sparse as $\mathbf{A}$, so their evaluation is slow, and their storage is the memory bottleneck for typical direct methods. Since APF does not compute the solution $\mathbf{X}$, such LU factors are not needed and can be dropped during the factorization. This means that APF is better than conventional direct methods even when only one input ($M = 1$) is considered.

APF is more efficient than computing selected entries of the Green's function $\mathbf{A}^{-1}$ (ref. [37]), which does not utilize the structure of equation (2). While advanced algorithms have been developed to exploit the sparsity of the inputs and the outputs during forward and backward substitutions[38] or through domain decomposition[39], they

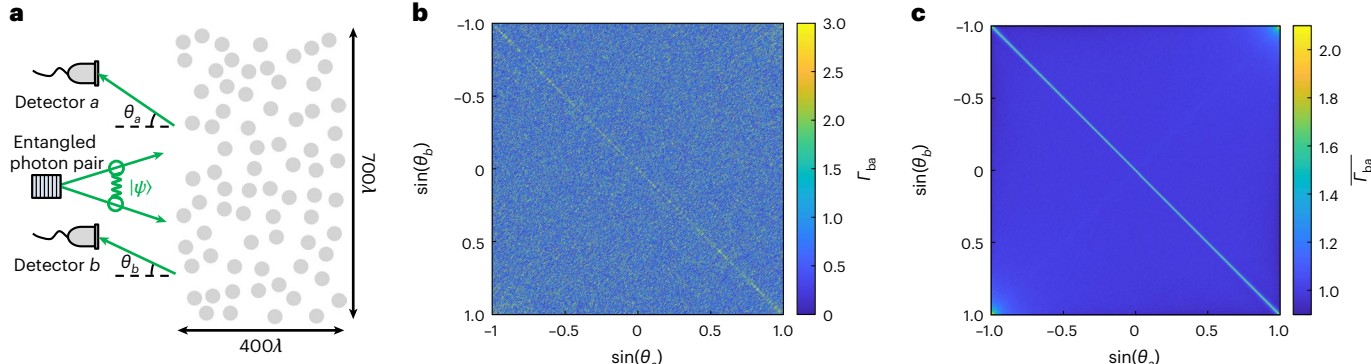

**Fig. 3 | Two-photon coherent backscattering from disorder. a,** Schematic of the system: maximally entangled photon pairs are reflected from a dynamic disordered medium, and the photon number correlation $\Gamma_{ba} = \langle\psi| : \hat{n}_b\,\hat{n}_a : |\psi\rangle$ of the two-photon wave function $|\psi\rangle$ is measured for pairs of reflected angles $\theta_a$ and $\theta_b$ for which the photon number operators are $\hat{n}_a$ and $\hat{n}_b$. This $\Gamma_{ba}$ is proportional to the square of the reflection matrix, as shown in equation (4). **b,c,** $\Gamma_{ba}$ for a single realization (**b**) and $\overline{\Gamma_{ba}}$ averaged over 4,000 realizations (**c**), normalized by $\overline{\Gamma_{ba}}$ away from the coherent backscattering peak.

still require an $\mathcal{O}(M)$ substitution stage, with a modest speed-up (a factor of 3 when $M$ is several thousand) and no memory usage reduction. APF is simpler yet much more efficient as it obviates the forward and backward substitution steps and the need for LU factors.

In most scenarios, the matrix **A** contains more nonzero elements than the matrices **B**, **C** and **S**, and we find the computing time and memory usage of APF to scale as $\mathcal{O}(N^{1.3})$ and $\mathcal{O}(N)$, respectively, in 2D (Supplementary Fig. 2), where $N = \text{nnz}(\mathbf{K})$ is the number of nonzero elements in the matrix **K** and is almost independent of $M$. When **B** and/or **C** contain more nonzero elements than **A**, we can compress matrices **B** and **C** through a data-sparse representation to reduce their numbers of nonzero elements to below that of **A**. For example, a plane-wave source spans a large area, but one can superimpose multiple plane-wave sources with a Fourier transform to make them spatially localized[8,9] and then truncate them with negligible error (Supplementary Sect. 5 and Supplementary Figs. 3 and 4).

Our implementation of APF is described in the Methods section and Supplementary Sects. 2 and 3, with pseudocodes shown in Supplementary Sect. 6.

Below, we consider two multi-channel systems while comparing the computing time, memory usage and accuracy of APF versus open-source electromagnetic solvers including a conventional finite-difference frequency-domain (FDFD) code named MaxwellFDFD using either (1) direct[40] or (2) iterative[41] methods, (3) an RGF code[42] and (4) an RCWA code named S4 (ref. [43]); see the Methods section for details. We do not include time-domain methods in the comparison since their iteration by time stepping is typically slower than an iterative frequency-domain solver[23]. We consider transverse magnetic polarization, starting with systems small enough for these solvers, then with larger problems that only APF can tackle.

## Large-scale disordered systems

Disordered systems are difficult to simulate given their large size-to-wavelength ratio, large number of channels, strong scattering and lack of symmetry. Here we consider one that is $W = 500\lambda$ wide and $L = 100\lambda$ thick, where $\lambda$ is the free-space wavelength, consisting of 30,000 cylindrical scatterers (Fig. 2a), discretized into 11.6 million pixels with a periodic boundary condition in $y$. On each of the $-x$ and $+x$ sides, $2W/\lambda = 1,000$ channels (plane waves with different angles) are necessary to specify the propagating components of an incident wavefront or outgoing wavefront at the Nyquist sampling rate (Supplementary Sect. 1A). So, we compute the scattering matrix with $M' = 2,000$ outputs and up to $M = 2,000$ inputs (including both sides).

It takes APF 3.3 min and 10 GiB of memory to compute the full scattering matrix; the other methods take 3,300–110,000,000 min using 7.0–1,200 GiB of memory for the same computation (Fig. 2b,c). The computing times of APF (with its breakdown shown in Fig. 2d), RGF and RCWA are all independent of $M$, though APF is orders of magnitude faster. MaxwellFDFD takes $\mathcal{O}(M)$ time due to its loop over the inputs. Reusing the LU factors helps, but the $M$ forward and backward substitutions take longer than factorization and become the bottleneck when $M \gtrsim 10$. Note that APF saves computing time and memory even in the single-input ($M = 1$) case.

The speed and memory advantage of APF grows further with the system size (Supplementary Fig. 5). Some of these solvers require more computing resources than we have access to, so their usage data (open symbols and grey-edged bars in Fig. 2b,c) are extrapolated based on smaller systems (Supplementary Fig. 5).

The relative $\ell^2$-norm error of APF due to numerical round-off is $10^{-12}$ here and grows slowly with an $\mathcal{O}(N^{1/2})$ scaling (Supplementary Fig. 6), while the iterative MaxwellFDFD method here has a relative $\ell^2$ error of $10^{-6}$.

Above, the matrices **B**, **C** and **S** all have fewer nonzero elements than the matrix **A** even for the largest $M$ at the Nyquist rate, so the APF computing time and memory usage are independent of $M$. Supplementary Sect. 9 and Supplementary Fig. 7 consider inputs and outputs placed in the interior of the disordered medium, where $M$ can grow larger. There, we observe that the APF computing time and memory usage stay constant until $M'M$ (the number of elements in **S**) grows beyond nnz(**A**) $\approx 5.8 \times 10^7$, above which they scale as $\mathcal{O}(M'M)$.

It was recently predicted that entangled photon pairs remain partially correlated even after multiple scattering from a dynamic disordered medium[44]. As an example, we demonstrate such two-photon coherent backscattering. Given a maximally entangled input state, the correlation between two photons reflected into directions $\theta_a$ and $\theta_b$ is[44]

$$\overline{\Gamma_{ba}} = \overline{\langle\psi| : \hat{n}_b\,\hat{n}_a : |\psi\rangle} \propto \overline{\left|(r^2)_{\theta_b,-\theta_a}\right|^2}, \tag{4}$$

where $|\psi\rangle$ is the two-photon wave function, $\hat{n}_a$ is the photon number operator in the reflected direction $\theta_a$, $:(...):$ stands for normal ordering, $r^2$ is the square of the medium's reflection matrix (that is, the scattering matrix with inputs and outputs on the same side) and the overbar indicates an ensemble average over disorder realizations. This requires the full reflection matrix with all incident angles and all outgoing angles, for many realizations, and the disordered medium must be wide (for angular resolution) and thick (to reach diffusive transport). Figure 3 shows

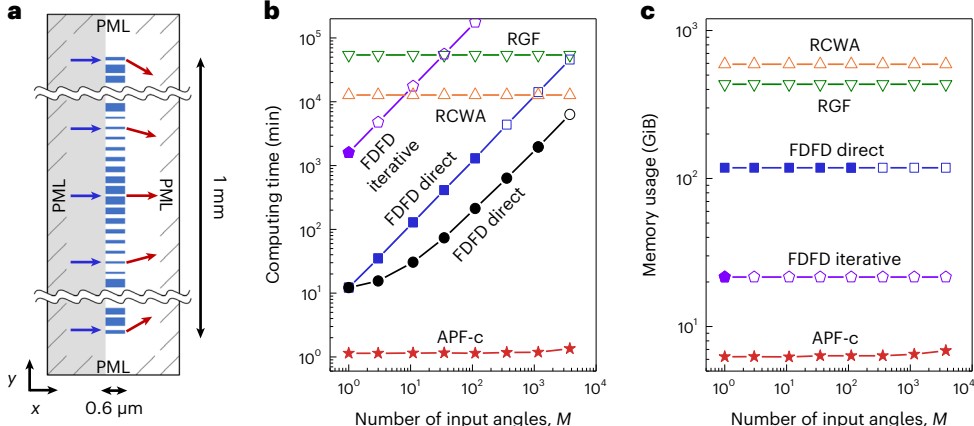

**Fig. 4 | Benchmarks on a large-area metasurface. a**, Schematic of the system: a 1-mm-wide metasurface consisting of 4,178 titanium dioxide ridges (blue rectangles) on a silica substrate (grey), operating at wavelength $\lambda = 532$ nm. PMLs are placed on all four sides. We compute the transmission matrix with up to $2W/\lambda = 3,761$ truncated plane-wave inputs from the left and with $M' = 2(W + 40\lambda)/\lambda = 3,841$ outgoing plane waves on the right. **b,c**, Computing time (**b**) and memory usage (**c**) versus the number $M$ of input angles using different methods. See the caption of Fig. 2 for details. APF-c denotes APF with the matrices **B** and **C** compressed.

the two-photon correlation function $\Gamma_{ba}$ computed using APF before and after averaging over 4,000 disorder realizations for a system that is $W = 700\lambda$ wide and $L = 400\lambda$ thick, consisting of 56,000 cylindrical scatterers, with a transport mean free path of $\ell_t = 9.5\lambda$. We find the correlation between photons reflected towards similar directions ($|\theta_b - \theta_a| \lesssim 0.1\lambda/\ell_t$) to be enhanced by a factor of 2. This demonstrates the existence of two-photon coherent backscattering in disordered media.

### Large-area metasurfaces

Metalenses are lenses made with metasurfaces[45]. When the numerical aperture (NA) is high, metalenses need to generate large phase gradients, so the variation from one unit cell to the next must be large, and the locally periodic approximation (LPA)[5,6] fails. Full-wave simulation remains the gold standard. Here, we consider metalenses with height of $L = 0.6$ μm and width of $W \approx 1$ mm, consisting of 4,178 unit cells of titanium dioxide ridges on a silica substrate (Fig. 4a), for a hyperbolic[46] phase profile with an NA of 0.86 and a quadratic[47] phase profile with an NA of 0.71 operating at wavelength $\lambda = 532$ nm (see Supplementary Sect. 10 and Supplementary Fig. 8 for details). Perfectly matched layers (PMLs) are placed on all sides, and the system is discretized with a grid size of $\Delta x = \lambda/40$ into over 11 million pixels. We compute the transmission matrix at the Nyquist sampling rate, with up to $M = 2W/\lambda = 3,761$ plane-wave inputs from the substrate side truncated within the width $W$ of the metalens (only considering angles that propagate in air), and sampling the transmitted field across a width $W_{out} = W + 40\lambda$ (to ensure that all the transmitted light is captured) projected onto $M' = 2W_{out}/\lambda = 3,841$ transmitted plane waves. Owing to the large aspect ratio of 1 mm to 0.6 μm, the number of nonzero elements in the matrices **B** and **C** is larger than that of **A**, so we compress **B** and **C** and denote this as APF-c (Supplementary Sect. 5).

It takes APF-c 1.3 min and 6.9 GiB of memory to compute this transmission matrix, while the other methods take 6,300–6,000,000 min using 22–600 GiB (Fig. 4b,c). Some of these values are extrapolated from smaller systems (Supplementary Fig. 9). Note that, even though RCWA is specialized for layered structures such as the metasurface considered here, the general-purpose APF-c still outperforms RCWA by 10,000 fold in speed and 87 fold in memory. The second-best solver here is MaxwellFDFD with the LU factors stored and reused, which takes 4,700 times longer while using 17 times more memory compared with APF-c.

The transmission matrix fully characterizes the metasurface's response to any input. Here, we use it with angular spectrum propagation (Supplementary Sect. 12) to obtain the complete angle dependence

of the exact transmitted profile (two profiles each shown in Fig. 5a,b; more shown in Supplementary Videos 1 and 2), the Strehl ratio and the transmission efficiency (Fig. 5c,d and Supplementary Sect. 13).

To quantify the accuracy of an approximation, we compute the relative $\ell^2$-norm error $\|\mathbf{I} - \mathbf{I}_0\|_2/\|\mathbf{I}_0\|_2$, with $\mathbf{I}_0$ being a vector containing the intensity at the focal plane within $|y| < W/2$ calculated from APF without compression, and **I** from an approximation. We consider two LPA formalisms: a standard one using the unit cells' propagating fields (LPA I) and one with the unit cells' evanescent fields included (LPA II) (Supplementary Sect. 14). LPA leads to errors up to 366% depending on the incident angle, with the angle-averaged error between 18% and 37% (Fig. 5e,f). Meanwhile, the compression errors of APF-c here average below 0.01% (Fig. 5e,f) and can be made arbitrarily small (Supplementary Fig. 10).

## Discussion

The APF method can enable a wide range of studies beyond the examples above. Full-wave simulations of imaging inside strongly scattering media[48] are now possible with APF. Inverse design using the adjoint method used to require $2M$ simulations given $M$ inputs[12]. With a suitable formulation, APF can consolidate the $2M$ simulations into a single or a few computations. Computing the thermal emission into a continuum[49] requires many simulations and can also be accelerated using APF. One may use APF to design classical and quantum photonic circuits with elements that couple numerous channels.

Beyond photonics, APF can be used for mapping the angle dependence of radar cross-sections, for microwave imaging[50], for full waveform inversion[51] and controlled-source electromagnetic surveys[38] in geophysics and for quantum transport simulations[52]. More generally, APF can efficiently evaluate matrices of the form $\mathbf{CA}^{-1}\mathbf{B}$ in numerical linear algebra, not limited to partial differential equations.

The present work performs partial factorization using MUMPS[33], for which the matrix **K** must be square. Therefore, we pad $M' - M$ columns to matrix **B** or $M - M'$ rows to matrix **C**, which is suboptimal when $M' \gg M$ (for example, when computing the field profile across a large volume for a small number of inputs) or $M' \ll M$. To efficiently handle these scenarios with APF, partial factorization that works with a rectangular **K** is desirable.

As the number of channels and the LU factor size are both much larger in three dimensions (3D), the advantage of APF over existing methods can potentially be greater in 3D than in 2D. In 3D, the memory usage due to the LU factors is the bottleneck for direct methods. Future

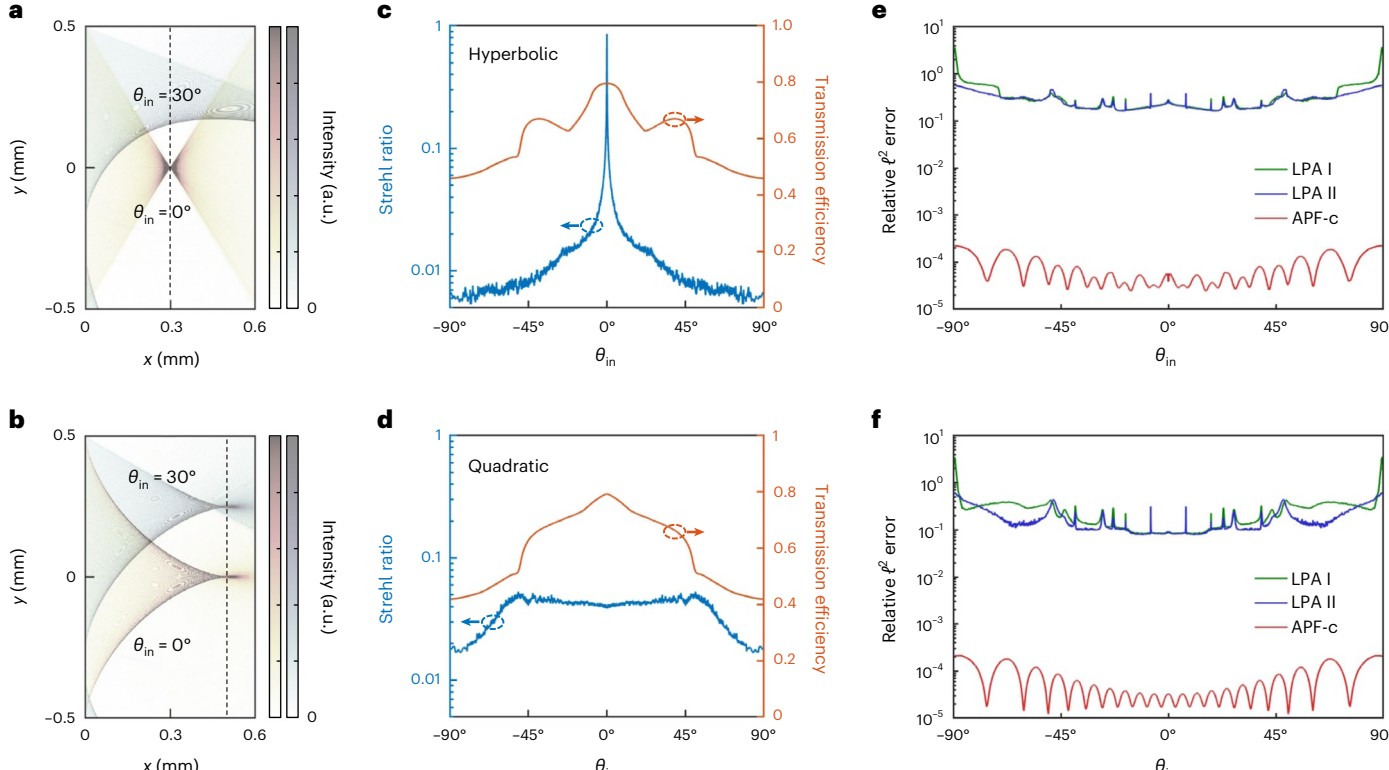

**Fig. 5 | All-angle full-wave characterization of millimetre-wide metalenses.** **a,b**, The intensity profile $|E_z(x,y)|^2$ of light after transmission through a hyperbolic metalens (**a**) or a quadratic metalens (**b**), for a plane wave incident from an angle $\theta_{\text{in}}$ of 0° and 30°. The intensity profiles for the two $\theta_{\text{in}}$ values are plotted with two different colour maps and overlaid with transparency, and the colour maps are saturated to show the low-intensity parts. The top of the

metalens ridges is at $x = 0$, and the dashed black lines indicate the focal plane. $\theta_{\text{in}} = \sin^{-1}(n_{\text{substrate}} \sin\theta_{\text{in}}^{\text{substrate}})$ is the incident angle in air. The complete dependence on $\theta_{\text{in}}$ is shown in Supplementary Videos 1 and 2. **c,d**, The full angle dependence of the Strehl ratio (**c**) and transmission efficiency (**d**). **e,f**, The errors on the focal-plane intensity from APF-c and from the LPA excluding (LPA I) and including (LPA II) the evanescent fields of the unit cells.

work could develop partial factorization schemes that minimize the temporary storage of such factors or even compute $\mathbf{CA}^{-1}\mathbf{B}$ without triangular factors. The expected usage of computing time and memory usage by APF in 3D follow that of the factorizing matrix $\mathbf{A}$, which is $\mathcal{O}(N^2)$ and $\mathcal{O}(N^{1.33})$, respectively, when using nested dissection ordering but could potentially be lowered by leveraging the low-rank property of the off-diagonal blocks[53]. APF-c can naturally work with overlapping-domain distribution strategies[7–9]. Multi-frontal parallelization can be used through existing packages such as MUMPS[33], and one may employ hardware accelerations with GPUs[9,54]. For systems with a small surface-to-volume ratio, it is also possible to apply APF to the boundary element method or $T$-matrix method, using the $\mathcal{H}$-matrix technique[27] for fast factorization.

## Methods

We implement APF under finite-difference discretization on the Yee grid in 2D (Supplementary Sect. 2) and compute the Schur complement using the MUMPS package[33] (version 5.4.1) with its built-in approximate minimum degree ordering. Outgoing boundaries are realized with PMLs[55]. We order the input/output channels and/or pad additional channels so that the matrix $\mathbf{K}$ is symmetric (Supplementary Sect. 3).

We use the same discretization scheme, same grid size and same subpixel smoothing[56] for the APF, MaxwellFDFD and RGF benchmarks. Numerical dispersion is not important for the disordered media example in Fig. 2, so we use a relatively coarse resolution of 15 pixels per $\lambda$ there. A finer resolution of 40 pixels per $\lambda = 532$ nm is used for the metasurface examples in Figs. 4 and 5 to have their transmission phase shifts accurate to within 0.1 rad (Supplementary Fig. 11).

In RGF[42], the outgoing boundary in the longitudinal direction is implemented exactly through the retarded Green's function of a semi-infinite discrete space[28]. For APF and MaxwellFDFD, one $\lambda$ of homogeneous space and 10 pixels of PML[55] are used to achieve an outgoing boundary with a sufficiently small discretization-induced reflection. The uniaxial PML is used in APF so that the matrix $\mathbf{A}$ is symmetric. The stretched-coordinate PML is used in MaxwellFDFD to lower the condition number[57].

For the MaxwellFDFD method with an iterative solver[41], we use its default biconjugate gradient method with its default convergence criterion of relative $\ell^2$ residual below $10^{-6}$. For the MaxwellFDFD method with a direct solver[40], we consider an unmodified version where the LU factors are not reused and a version modified to have the LU factors stored in memory and reused for the different inputs.

For the RCWA simulations, we use its default closed-form Fourier-transform formalism implemented in S4 (ref. [43]). For the example in Fig. 4, we use a single layer with five Fourier components per unit cell where the cell width is 239 nm (that is, 11 Fourier components per $\lambda$), which gives accuracy comparable to APF, MaxwellFDFD and RGF (Supplementary Fig. 11). For the example in Fig. 2, we use 15 layers per $\lambda$ axially (the same as the discretization grid size used in the other methods) with 4.1 Fourier components per $\lambda$ laterally (by scaling it down in proportion to the reduced spatial resolution in APF, MaxwellFDFD and RGF).

Note that the RGF[42] and S4 (ref. [43]) codes do not support an outgoing boundary in the transverse $y$ direction. The computing time and memory usage for RGF and S4 in Fig. 4 are extrapolated from simulations on smaller systems adopting a periodic transverse boundary (Supplementary Fig. 9). To simulate the example in Fig. 4 using RGF or

S4, one needs to additionally implement PML in the *y* direction and to further increase the system width. Doing so will slightly increase their computing time and memory usage, which we disregard.

All the computing time and memory usage values are obtained from computations using a single core without parallelization on identical Intel Xeon Gold 6130 nodes on the USC Center for Advanced Research Computing's Discovery cluster with 184 GiB of memory available per node.

## Data availability

Numerical source data for Figs. 2, 4 and 5c–f are available with this manuscript in the Source Data section. Numerical source data for Figs. 3b–c and 5a,b are available on Zenodo[58]. All data in this study are generated by running our code[59], MaxwellFDFD[40,41] and S4 (ref. [43]).

## Code availability

We implement the APF method and the RGF method within our software Maxwell's Equations Solver with Thousands of Inputs (MESTI). The code, documentation and examples are available on GitHub[59] under the GPL-3.0 license. MESTI supports both polarizations, all common boundary conditions, real or complex frequencies, with any permittivity profile and any list of input source profiles and output projection profiles (user specified or automatically built). The specific version of MESTI used to produce the results in this manuscript is also available on Zenodo[60].

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

## Acknowledgements

We thank Y. Bromberg, M. Safadi, A. Goetschy, C. Sideris, A. D. Stone, S. G. Johnson, S. Li and M. Torfeh for useful discussions. This work is supported by the National Science Foundation CAREER award ECCS-2146021 and the Sony Research Award Program. Computing resources are provided by the Center for Advanced Research Computing (CARC) at the University of Southern California.

## Author contributions

H.-C.L. and C.W.H. performed the simulations and data analysis. C.W.H., H.-C.L. and Z.W. wrote the APF codes. C.W.H. developed the theory and supervised the research. All authors contributed to designing the systems, discussing the results and preparing the manuscript.

## Competing interests

The authors declare no competing interests.

## Additional information

**Correspondence and requests for materials** should be addressed to Chia Wei Hsu.

