## [Peer Review File · Nature Computational Science]

Peer Review Information

Journal: Nature Computational Science

Manuscript Title: Fast multi-source nanophotonic simulations using augmented partial factorization

Corresponding author name(s): Chia Wei Hsu Click here to enter text.

Reviewer Comments & Decisions:

Decision Letter, initial version:

Dear Professor Hsu,

Your manuscript "Fast multi-source nanophotonic simulations using augmented partial factorization" has now been seen by 2 referees (referee #2 unfortunately cannot help us review the paper anymore), whose comments are appended below. You will see that while they find your work of interest, they have raised points that need to be addressed before we can make a decision on publication.

The referees' reports seem to be quite clear. Naturally, we will need you to address all of the points raised.

While we ask you to address all of the points raised, the following points need to be substantially worked on:

- Please add discussions about the potential limitations.
- Please provide clarifications and explanations about those technical points raised by our referees.
- Please improve the accessibility of your manuscript to a broad audience.
- Please add experimental data to validate your simulation findings.

Best regards,

Jie Pan, Ph.D.
Associate Editor
Nature Computational Science

Reviewers comments:

Reviewer #1 (Remarks to the Author):

The manuscript presents a fast and efficiency approach to calculate the scattering matrix of large-scale photonic systems. Their approach is based on the similarity between the general expression of scattering matrix and Schur complement. The authors propose a single factorization scheme to calculate the Schur complement efficiently.

The contribution of this work, in my opinion, is substantial to the photonics community, in which the lack of fast and efficient full-wave numerical solvers has become the bottleneck for exploration and design of large photonic systems. Hardware acceleration such as GPU parallelization can drastically reduce the simulation time, but a large number of high-performance hardware will be required as simulating large photonic systems require tremendous memory. An idea solver thus should be fast and memory-efficient. In this regard, this work is one of the few works that are addressing this challenge, and the proposed scheme is impressively fast and efficient. I commend the authors for their excellent work.

Overall, I recommend this work for publication. My specific technical comments are listed below:

1. Line 20-21: coupled-mode theory (CMT) is not limited to a few resonances. Numerically, the number of resonances in CMT is also limited by the available memory; it may not be a large number, but it is certainly not "a few resonances".
2. Frequency spectrum is also one of the physical quantities of great interest. A short discussion regarding how the APF approach can be utilized to calculate frequency spectrum will be helpful.
3. Is the scaling of computing time and memory usage for 3D structures the same as 2D structures?

Reviewer #3 (Remarks to the Author):

The authors report a fast method to calculate the full-wave solution of Maxwell's equations, called "augmented partial factorization (APF)". The method is shown to outperform other full-wave numerical solvers of Maxwell's equations such as the FDFD, RCWA and RGF in saving the computation time and memory, as shown by the numerical results in Figs. 2 and 4. Two numerical examples of large-scale disordered scatterers (Figs. 2 and 3) and large-area metalens (Figs. 4 and 5) are provided to illustrate the utility of the method. As explained in Section I, the proposed APF is generally applicable to any system whose output has a linear dependence on the input as described by the general matrix Eq. (2), and its computational efficiency comes from the almost independence of the computation time with respect to the number M of input channels. In view of the general applicability and the great computational advantages of the proposed APF, I recommend publication of the manuscript provided that the following issues are addressed properly.

1. Despite the great advantages of the proposed APF, I believe that there should exist some deficiencies or limitations of the APF, which should be clarified for the sake of an objective evaluation of the APF in comparison with other full-wave numerical solvers of Maxwell's equations. For example, if calculating the electromagnetic near field within the inhomogeneous region of photonic nanostructures (instead of calculating the far field in the homogeneous free space as provided in Figs. 5a and b or the Supplementary Movies S1 and S2), which is routinely required in nanophotonics researches, can the APF outperform other full-wave numerical solvers?
2. The proposed APF relies on the key Eq. (3), which can derive $S=-H$ after simple algebraic operations. Then one can obtain the required generalized scattering matrix S from H which is calculated numerically via the LU factorization of matrix K as described by the second equality in Eq. (3). This process shows that if changing the matrix B that represents the inputs of the system, a repetitive LU factorization of K is required to obtain H . In comparison, for some other numerical solvers such as the "FDFD direct" shown by the black circles in Fig. 2b, the LU factors are reused when changing B by increasing the number M of input angles. Now the following question may need answer or further explanations. Because the LU factorization represents the major computational amount for both the APF and the "FDFD direct", why the APF with a repetitive LU factorization can be faster than the "FDFD direct" without a repetitive LU factorization?
3. In Eq. (2), the necessity or advantages of introducing matrix D to remove the baseline contribution may need further explanation.
4. As mentioned after Eq. (3) and also at the end of the manuscript, the LU factors do not need to be triangular. Does this mean that the LU factors are triangular for the present numerical implementation of APF? This point should be stated clearly. I guess that for the LU factorization of K in Eq. (3), non-triangular LU factors should need distinctly less computational amount than triangular LU factors, which is also implied by the authors at the end of the manuscript. I suggest the authors to add more comments or relevant references (if exist) on further reducing the computational amount by adopting non-triangular LU factors.
5. In Fig. 2a, the coordinates can be provided.

Author Rebuttal to Initial comments

Authors' Response to Reviewer 1

General Comments. The manuscript presents a fast and efficiency approach to calculate the scattering matrix of large-scale photonic systems. Their approach is based on the similarity between the general expression of scattering matrix and Schur complement. The authors propose a single factorization scheme to calculate the Schur complement efficiently.

The contribution of this work, in my opinion, is substantial to the photonics community, in which the lack of fast and efficient full-wave numerical solvers has become the bottleneck for exploration and design of large photonic systems. Hardware acceleration such as GPU parallelization can drastically reduce the simulation time, but a large number of high-performance hardware will be required as simulating large photonic systems require tremendous memory. An idea solver thus should be fast and memory-efficient. In this regard, this work is one of the few works that are addressing this challenge, and the proposed scheme is impressively fast and efficient. I commend the authors for their excellent work.

Overall, I recommend this work for publication. My specific technical comments are listed below.

Response: We thank the reviewer for the high regard of this work and the technical comments that help improve this paper.

Comment 1-1

Line 20-21: coupled-mode theory (CMT) is not limited to a few resonances. Numerically, the number of resonances in CMT is also limited by the available memory; it may not be a large number, but it is certainly not “a few resonances”.

Response: This sentence is indeed not clear, and we thank the reviewer for pointing it out. We meant to say that each resonator should contain few resonances (to avoid spectral overlap and to ease the mode computation), but indeed many such resonators can be coupled so the total number of resonances can be large. We have changed “coupled-mode theory is restricted to systems with few isolated resonances” to

coupled-mode theory requires systems with isolated resonances³⁻⁵.

At the end of this sentence, we also added Ref. 5 (previously Ref. 49), where the requirements of CMT are studied in detail.

Comment 1-2

Frequency spectrum is also one of the physical quantities of great interest. A short discussion regarding how the APF approach can be utilized to calculate frequency spectrum will be helpful.

Response: Here we use APF in frequency domain. So, like other frequency-domain methods, computation at different frequencies can be done independently (e.g., on different machines, or on different cores of the same machine, in an “embarrassingly parallel” manner.) We revised the following sentence on lines 197–200 of page 3 of the main text to comment on this:

As a frequency-domain method, it allows arbitrary material dispersion, and the response at different frequencies can be computed independently in parallel or in series.

Comment 1-3

Is the scaling of computing time and memory usage for 3D structures the same as 2D structures?

Response: The scaling in 3D is nontrivial. In 2D, we found that systems with very different sizes and aspect ratios and with different matrix-reordering algorithms follow roughly the same scaling for computing time and memory usage when we consider the number of nonzero elements $N = \text{nnz}(\mathbf{K})$ as the “size parameter” (as shown in Supplementary Fig. S1, which becomes Fig. S2 in the revised version). However, we just recently finished writing an initial version of APF solver for 3D vectorial Maxwell’s equations, and very preliminary results show that the computing time and memory usage in 3D depend on more variables—for the same $\text{nnz}(\mathbf{K})$, the computing time and memory usage can change significantly given different aspect ratios of the $x/y/z$ dimensions, different matrix-reordering algorithms used (the AMD ordering we use in 2D is no longer efficient in 3D), and even different boundary conditions. This is likely due to the Maxwell-operator matrix \mathbf{A} having a more connected sparsity pattern. For cube-shaped geometries with METIS ordering, we observe $\mathcal{O}(N^2)$ scaling for computing time and $\mathcal{O}(N^{1.3})$ scaling for memory usage when the block low-rank format is not used. When the block low-rank format is used, we expect $\mathcal{O}(N^{1.76})$ scaling for computing time and $\mathcal{O}(N^{1.17} \log(N))$ scaling for memory usage (per Table 6 and Fig. 10 of Ref. 83). When the geometry is 2D-like (e.g., a thin metasurface), we observe scaling closer to that in 2D.

We revised the following sentence on lines 450-453 in the Discussion section of the main text:

The expected APF computing time in 3D follows that of factorizing matrix \mathbf{A} , with $\mathcal{O}(N^2)$ and $\mathcal{O}(N^{1.76})$ scaling for cube-shaped geometries without and with the use of block low-rank format respectively⁸³.

Authors' Response to Reviewer 3

General Comments. The authors report a fast method to calculate the full-wave solution of Maxwell's equations, called "augmented partial factorization (APF)". The method is shown to outperform other full-wave numerical solvers of Maxwell's equations such as the FDFD, RCWA and RGF in saving the computation time and memory, as shown by the numerical results in Figs. 2 and 4. Two numerical examples of large-scale disordered scatterers (Figs. 2 and 3) and large-area metalens (Figs. 4 and 5) are provided to illustrate the utility of the method. As explained in Section I, the proposed APF is generally applicable to any system whose output has a linear dependence on the input as described by the general matrix Eq. (2), and its computational efficiency comes from the almost independence of the computation time with respect to the number M of input channels. In view of the general applicability and the great computational advantages of the proposed APF, I recommend publication of the manuscript provided that the following issues are addressed properly.

Response: We thank the reviewer for the recommendation and the comments that help improve this paper.

Comment 3-1

Despite the great advantages of the proposed APF, I believe that there should exist some deficiencies or limitations of the APF, which should be clarified for the sake of an objective evaluation of the APF in comparison with other full-wave numerical solvers of Maxwell's equations. For example, if calculating the electromagnetic near field within the inhomogeneous region of photonic nanostructures (instead of calculating the far field in the homogeneous free space as provided in Figs. 5a and b or the Supplementary Movies S1 and S2), which is routinely required in nanophotonics researches, can the APF outperform other full-wave numerical solvers?

Response: We thank the reviewer for bringing up this important point. Indeed, there are two situations when APF in its present implementation may not be desirable. The first situation is when the number of outputs M' greatly exceeds the number of inputs M (e.g., computing the near-field profile across a large volume for a small number of inputs). In theory, the APF approach as in Eq. (3) of the main text still applies when $M' \gg M$. But in practice, we use the MUMPS package to perform the partial

factorization, and MUMPS currently requires such matrix \mathbf{K} to be square, so we must pad $M' - M$ redundant columns to the input matrix \mathbf{B} in order to use MUMPS. Since the APF advantage is more significant when M is large, the redundancy from such padding and the subsequent partial factorization can more than offset the benefit of APF when M is small. One can overcome this limitation by allowing partial factorization on a rectangular matrix \mathbf{K} , and we plan to work with the MUMPS developers in the future to add this functionality. To address this limitation and its solution, we added the following paragraph on lines 439–446 in the Discussion section of the main text:

The present work performs partial factorization using MUMPS, for which matrix \mathbf{K} must be square. This requires us to pad $M' - M$ columns to matrix \mathbf{B} or $M - M'$ rows to matrix \mathbf{C} , which is suboptimal when $M' \gg M$ (e.g., computing the field profile across a large volume for a small number of inputs) or $M' \ll M$. Partial factorization implementations that work with rectangular \mathbf{K} are desirable.

The second situation is when the generalized scattering matrix \mathbf{S} is so large that it contains much more elements than $N = \text{nnz}(\mathbf{A})$. In such scenario, APF is still very efficient, but its memory usage will increase. Since matrix \mathbf{S} already contains $M'M$ elements, computing these elements and storing them must take at least $\mathcal{O}(M'M)$ time and memory even in the best-case scenario, regardless of the method used. So, in the limit of M becoming very large, the M independence of APF computing time and memory usage must cease to hold. When only the far-field quantities are of interest, one has at most two channels per wavelength of the surface (Nyquist sampling rate), so the number of input/output channels can be large but has an upper bound; APF is efficient enough that we do not observe the $\mathcal{O}(M'M)$ scaling even at the largest possible M in such case. But for near-field computations with inputs/outputs inside the inhomogeneous region, the number of inputs/outputs can potentially be much larger, for which the inevitable $\mathcal{O}(M'M)$ scaling emerges.

To study such scenario, we consider the disordered structure in Fig. 2 of the main text, but place the input sources and output projections at M points in the interior of the scattering medium instead. The generalized scattering matrix we compute in this case is therefore the retarded Green's function evaluated at such points. We use the same detection points as the source points (so, $M' = M$), with these points evenly spaced inside the medium; the spacing is decreased progressively to raise M up to $M = 44,000$ (upper limit with our computing resources). Figure R1 below shows the computing time and memory usage of APF and MaxwellFDFD. The red dotted lines show fittings to a second-order polynomial in M . We see that the APF computing time and memory usage remain constant up to $M \sim 7,000$, beyond which we observe an $\mathcal{O}(M'M) = \mathcal{O}(M^2)$ scaling, which is the best possible scaling at large M . MaxwellFDFD is much slower, and its memory usage also scales as $\mathcal{O}(M'M)$ for large M since that is how much memory

it takes just to store \mathbf{S} , but it has a smaller prefactor. By extrapolating the curves, we expect APF to use more memory than MaxwellFDFD when $M \gtrsim 60,000$. But, APF is still estimated to be over 4,000 times faster than MaxwellFDFD (even with the LU factors reused) at $M = 60,000$.

Fig. R1. Timing and memory usage for computing the Green’s function.

Computing time (a) and memory usage (b) versus the number of spatial points when computing the retarded Green’s function $G(\mathbf{r}, \mathbf{r}')$ at M source points \mathbf{r}' and $M' = M$ detection points \mathbf{r} inside the disordered system in Fig. 2 of the main text. Red dotted lines show fittings to a second-order polynomial in M , and black dashed lines indicate $M_0 = \sqrt{\text{nnz}(\mathbf{A})} \approx 7,600$, above which the computed Green’s function has more data points than $\text{nnz}(\mathbf{A})$. The two “FDFD direct” curves correspond to an unmodified version of MaxwellFDFD (blue squares) and one modified to have the LU factors reused for different inputs (black circles), with open symbols being estimations from smaller M .

Intuitively, we expect the $\mathcal{O}(M'M)$ scaling to take over when matrix \mathbf{S} contains more elements than $N = \text{nnz}(\mathbf{A})$. With $M' = M$ in this example, we expect the transition to take place near $M_0 = \sqrt{N} \approx 7,600$. This M_0 , shown as black dashed lines in Fig. R1, closely predicts the occurrence of the transition.

To show these results, we added a new Section 9 to the Supplementary Materials, with the Fig. R1 above included as Fig. S7. In the main text, we added “and \mathbf{S} ” on line 225 of page 4:

In most applications, matrix \mathbf{A} contains more nonzero elements than matrices \mathbf{B} , \mathbf{C} , and \mathbf{S} , and we find in such scenarios that...

and the following paragraph on lines 304–313 of page 5:

Above, matrices \mathbf{B} , \mathbf{C} , and \mathbf{S} all have less nonzero elements than matrix \mathbf{A} even for the largest M at the Nyquist rate, so the APF computing time and memory usage is independent of M . Supplementary Sec. 9 and Fig. S7 consider inputs and outputs placed in the interior of the disordered medium, where M can grow larger. There, we observe the APF computing time and memory usage to stay constant until $M'M$ (the number of elements in \mathbf{S}) grows beyond $N = \text{nnz}(\mathbf{A}) \approx 5.8 \times 10^7$, above which they follow the optimal $\mathcal{O}(M'M)$ scaling.

Comment 3-2

The proposed APF relies on the key Eq. (3), which can derive $\mathbf{S} = -\mathbf{H}$ after simple algebraic operations. Then one can obtain the required generalized scattering matrix \mathbf{S} from \mathbf{H} which is calculated numerically via the LU factorization of matrix \mathbf{K} as described by the second equality in Eq. (3). This process shows that if changing the matrix \mathbf{B} that represents the inputs of the system, a repetitive LU factorization of \mathbf{K} is required to obtain \mathbf{H} . In comparison, for some other numerical solvers such as the “FDFD direct” shown by the black circles in Fig. 2b, the LU factors are reused when changing \mathbf{B} by increasing the number M of input angles. Now the following question may need answer or further explanations. Because the LU factorization represents the major computational amount for both the APF and the “FDFD direct”, why the APF with a repetitive LU factorization can be faster than the “FDFD direct” without a repetitive LU factorization?

Response: If a different input matrix \mathbf{B}_2 is needed, those inputs can be appended to the original set of inputs \mathbf{B}_1 , as $\mathbf{B} = [\mathbf{B}_1, \mathbf{B}_2]$, and computed during the original APF computation. There is no need for a new APF. There are applications where the next input of interest is determined by results from the previous input(s) considered; since any arbitrary input can be written as the superposition of a set of “input basis,” in such scenario one can simply use one APF to compute the full scattering matrix containing the entire input basis set, and then use such scattering matrix to obtain the outputs given any arbitrary input. In general, with APF, one would only perform a single partial factorization regardless of how many inputs there are and whether they change or not. To make this clearer, we added the following sentence on lines 139–141 of the main text (page 3):

Given the generalized scattering matrix \mathbf{S} , the response to other inputs can be obtained from superposition, as in Eq. (1).

and the following sentence on lines 187–188:

and repetitions over different inputs are no longer necessary.

We also note that with the “FDFD direct” shown by the black circles in Fig. 2b, the LU factorization is computed only once, but M forward and backward substitutions using the triangular factors are also needed in addition. Those forward and backward substitutions dominate the overall computing time when $M > 10$, which becomes the main reason it is much slower than APF. We show the computing time breakdown of that “FDFD direct” curve in Fig. R2 below.

Fig. R2. The computing time breakdown of the black “FDFD direct” curve in Fig. 2b of the main text. “Triangular solve” refers to the M forward and backward substitutions using the LU factors.

To make this clearer, we revised the following sentence on lines 289–291 (page 5) of the main text:

the M forward and backward substitutions take longer than factorization and become the bottleneck when $M \gtrsim 10$.

Comment 3-3

In Eq. (2), the necessity or advantages of introducing matrix \mathbf{D} to remove the baseline contribution may need further explanation.

Response: Indeed this should have been explained better. In a scattering problem, the total field is

$$E^{\text{tot}}(\mathbf{r}) = E^{\text{in}}(\mathbf{r}) + E^{\text{Sca}}(\mathbf{r}), \quad (\text{R1})$$

where $E^{\text{in}}(\mathbf{r})$ is the incident field (which is incoming on the incident side but outgoing on the transmitted side), and $E^{\text{Sca}}(\mathbf{r})$ is the scattered field. Reflection and transmission are defined by the outgoing components of the total field $E^{\text{tot}}(\mathbf{r})$, which means that reflection is defined only by the scattered field $E^{\text{Sca}}(\mathbf{r})$ on the incident side, while transmission is defined by the total field $E^{\text{tot}}(\mathbf{r}) = E^{\text{in}}(\mathbf{r}) + E^{\text{Sca}}(\mathbf{r})$ on the transmitted side. In this work, we use a surface source as \mathbf{B} to generate the incident field, and the resulting $\mathbf{A}^{-1}\mathbf{B}$ is the total field $E^{\text{tot}}(\mathbf{r})$ (except on the other side of the source), so $\mathbf{C}\mathbf{A}^{-1}\mathbf{B}$ is the projection of the total field $E^{\text{tot}}(\mathbf{r})$. But reflection should be defined by the projection of $E^{\text{Sca}}(\mathbf{r})$ instead, so we need to subtract the projection of $E^{\text{in}}(\mathbf{r})$ to obtain reflection; that subtraction is \mathbf{D} and is the δ_{ba} in Eq. (S11a) and Eq. (S24a). Transmission is the projection of $E^{\text{tot}}(\mathbf{r})$, so no subtraction is needed, and there is no δ_{ba} in Eq. (S11b) and Eq. (S24b). This is illustrated by Fig. R3 below.

Fig. R3. Schematic illustration of $\mathbf{S} = \mathbf{C}\mathbf{A}^{-1}\mathbf{B} - \mathbf{D}$ in Eq. (S11) and Eq. (S24). Given a surface source, the solution $\mathbf{A}^{-1}\mathbf{B}$ at $x \geq 0$ is the total field $E = E^{\text{in}} + E^{\text{Sca}}$. Projecting such total field at $x = L$ onto the transverse modes through multiplication with matrix \mathbf{C} gives the transmission matrix. Projecting the total field at $x = 0$ gives the reflection matrix (from the projection of E^{Sca}) plus \mathbf{D} (from the projection of E^{in}), so \mathbf{D} is subtracted to yield the reflection matrix.

In the supplementary materials, we added this figure as Fig. S1 and revised the

paragraph under Eq. (S11) for better clarity. In the main text, we revised the following sentence on lines 129–131 of page 2:

Matrix $\mathbf{D} = \mathbf{C}\mathbf{A}_0^{-1}\mathbf{B} - \mathbf{S}_0$ subtracts the baseline contribution from the incident field (Supplementary Fig. S1).

and added the following sentence on lines 162–163 of page 3:

In this particular example, $\mathbf{D} = \mathbf{I}$ is the identity matrix

Note that what \mathbf{D} is depends on how the source profile \mathbf{B} is set up. One can also solve scattering problems using a volume source (instead of surface source) as \mathbf{B} , for which the source $b = -\mathbf{A}E^{in}$ is non-zero across the whole volume where $\mathbf{A} \neq \mathbf{A}_0$. In this case, $\mathbf{A}^{-1}\mathbf{B}$ is the scattered field $E^{Sca}(\mathbf{r})$ (instead of the total field), and $\mathbf{C}\mathbf{A}^{-1}\mathbf{B}$ is the projection of the scattered field. Here, we need to use \mathbf{D} to add the projection of $E^{in}(\mathbf{r})$ to get transmission, while reflection does not need such \mathbf{D} .

In Eq. (2) of the main text, we want to introduce a general formalism that applies to any problem (can be any linear response in any form, not just reflection and transmission) and is independent of how one chooses to set up the the source. The expression $\mathbf{D} = \mathbf{C}\mathbf{A}_0^{-1}\mathbf{B} - \mathbf{S}_0$ is not as intuitive as the Fig. R3 above but is the most general expression, because it ensures that the generalized scattering matrix \mathbf{S} reduces to the known reference \mathbf{S}_0 when the system \mathbf{A} is reduced to a reference system \mathbf{A}_0 . (Such a reference system can be vacuum, or any system of convenience.) To make this clearer, we added the following sentence on lines 133–134 of the main text:

it ensures that \mathbf{S} reduces to \mathbf{S}_0 when \mathbf{A} becomes \mathbf{A}_0 .

Comment 3-4

As mentioned after Eq. (3) and also at the end of the manuscript, the LU factors do not need to be triangular. Does this mean that the LU factors are triangular for the present numerical implementation of APF? This point should be stated clearly. I guess that for the LU factorization of \mathbf{K} in Eq. (3), non-triangular LU factors should need distinctly less computational amount than triangular LU factors, which is also implied by the authors at the end of the manuscript. I suggest the authors to add more comments or relevant references (if exist) on further reducing the computational amount by adopting non-triangular LU factors.

Response: Yes, while the APF formalism of Eq. (3) does not require the \mathbf{L} and \mathbf{U} matrices to be triangular, the implementation in this paper still uses triangular LU factors (which are computed but not stored in memory) because that is what the MUMPS package does.

Indeed, we expect that APF computations can be further accelerated when non-triangular factors are adopted, or more generally when new partial factorization algorithms are developed with the goal of computing the Schur complement but not the LU factors. We are not aware of prior work that does so, since existing factorization methods are mostly designed to solve conventional systems of linear equations where the LU factors must be triangular. This is one of the future directions we plan to pursue.

To clarify and to comment on the above, we added the following sentence on lines 241–242 (page 4) of the main text (which was previously mentioned in the Methods section):

with the Schur complement computed using the MUMPS package.

and revised the last sentence (lines 463–469) of the main text as:

Existing factorization methods were developed with the goal of using the triangular LU factors, but APF introduces a class of applications where such factors in Eq. (3) are not needed and do not need to be triangular; future work can develop partial factorization methods that compute $\mathbf{CA}^{-1}\mathbf{B}$ with non-triangular factors for further efficiency gain.

Comment 3-5

In Fig. 2a, the coordinates can be provided.

Response: We thank the reviewer for the suggestion and have added the (x, y) coordinates to Fig. 2a.

Decision Letter, first revision:

Dear Dr. Hsu,

Thank you for submitting your revised manuscript "Fast multi-source nanophotonic simulations using augmented partial factorization" (NATCOMPUTSCI-22-0660A). It has now been seen by the original referees and their comments are below. The reviewers find that the paper has improved in revision, and therefore we'll be happy in principle to publish it in Nature Computational Science, pending minor revisions to satisfy the referees' final requests and to comply with our editorial and formatting guidelines.

Thank you again for your interest in Nature Computational Science Please do not hesitate to contact me if you have any questions.

Sincerely,

Jie Pan, Ph.D.
Associate Editor
Nature Computational Science

Reviewer Comments

Reviewer #1 (Remarks to the Author):

The authors have addressed all reviewer's comments very well. I recommend it for publication.

Reviewer #3 (Remarks to the Author):

The authors have made sufficient improvement of the manuscript in reponse to the issues raised in my previous report, with detailed explanations and supporting data. I recommend the present form of the manuscript for publication.

Final Decision Letter:

Dear Professor Hsu,

We are pleased to inform you that your Article "Fast multi-source nanophotonic simulations using augmented partial factorization" has now been accepted for publication in Nature Computational Science.

Once your manuscript is typeset, you will receive an email with a link to choose the appropriate publishing options for your paper and our Author Services team will be in touch regarding any additional information that may be required.

Best regards,

Jie Pan, Ph.D.

Associate Editor
Nature Computational Science